# Elucidating and Expanding the Restorative Theory Framework to Comprehend Influential Factors Supporting Ageing-in-Place: A Scoping Review

**DOI:** 10.3390/ijerph20186801

**Published:** 2023-09-21

**Authors:** Anne Johanna Jacoba Grave, Louis Neven, Masi Mohammadi

**Affiliations:** 1Smart Architectural Technologies, Department of the Built Environment, Eindhoven University of Technology, Groene Loper 3, 5612 AE Eindhoven, The Netherlands; m.mohammadi@tue.nl; 2Research Group Architecture in Health, HAN University of Applied Sciences, Ruitenberglaan 26, 6826 CC Arnhem, The Netherlands; louis.neven@han.nl

**Keywords:** psychological restoration, older adults, restorative environments, mental health, ageing-in-place

## Abstract

Exposure to stress and attention fatigue resulting from changes in capabilities and residing in environments that do not align with individual needs can adversely impact older adults’ mental health and complicate ageing-in-place. Research into the psychological restoration process can help assist in alleviating these issues. Existing research on restoration perspectives has predominantly centred on university students and lacks comprehensive insights into older adults. Consequently, this study seeks to acquire a deeper understanding of the restorative theory framework within the context of ageing populations. We identified and analysed thirty-nine papers on the restoration process of older adults employing the scoping review method. Our findings indicate that adjustments to the general restorative theory framework are imperative for ageing populations. By incorporating additional features—such as being with and familiarity—the framework can more effectively support the development of age-inclusive neighbourhoods that enhance the mental health of the older population and facilitate healthy ageing-in-place. While more in-depth research is required on the restoration process of older adults, this research marks the initial in adapting the general framework to ageing populations. Furthermore, insight is given into how the adapted framework can contribute to help address the challenges of global ageing and support ageing-in-place.

## 1. Introduction

Due to scientific advances, improved living conditions and medical breakthroughs, our global population is rapidly ageing. In 2050, the global population of people aged 60 and older is expected to double to nearly 2.1 billion people [1]. An ageing population affects society in many ways, including public policies, health care and social services, as well as how to design our environment [2,3]. To this end, the World Health Organization launched its Ageing and health strategy [4], which aims to support the older population through “healthy ageing”. This policy also ties in with the wish of many older adults to age in place and grow old in a familiar living environment [3,5]. Unfortunately, for many older adults, ageing-in-place becomes difficult when a misfit between their living environment and their changing capabilities arises [6,7]. As a result of the ageing process, older adults can experience changing needs and capabilities, for example, declined physical strength, self-care abilities and increased risk of developing chronic conditions and mental health issues [8,9,10,11]. These changes in the individual let people experience their living environment differently. To continue living at home, older adults must adapt to cope with the environment [6].

The literature highlights three pathways that individuals follow when adapting to changes in their needs and capabilities in response to environmental shifts: (1) the stress perspective: this pathway focuses on mitigating heavy demands; (2) the coping perspective: this approach seeks to enhance the availability of resources for adaptation; and (3) the restoration perspective: within this context, the restoration perspective aims to provide opportunities to recover resources. This includes activities like attention restoration and psychophysiological stress recovery [12,13,14]. The restoration process is defined as *“the process of recovering physiological, psychological and social resources that have become diminished in efforts to meet the demands of everyday life”* [15] (p. 164). In the field of research on ageing-in-place, emphasis is often put on the first two pathways. Researchers frequently investigate strategies to reduce exposure to environmental stressors, such as air pollution and noise, promote physical activity and health through the design of therapeutic or biophilic environments and foster social cohesion (e.g., [10,16,17,18,19]). However, studies focusing on the third pathway—psychological restoration in the context of the older population—remain relatively sparse [17,20]. Existing research on the restoration perspective primarily focuses on university students. While older adults do require restoration of physiological, psychological and social resources [17,20], they are more susceptible to attention fatigue and life stressors, as evidenced by previous research [21,22]. This susceptibility is not solely attributed to age-related changes in capabilities; it is also linked to a heightened likelihood of encountering stressful life events. These events may include declines in socioeconomic status, alterations in social structures and shifts in family support dynamics [11,23,24]. If older adults in their environment do not have enough opportunities to restore resources, attention fatigue and chronic stress can arise, affecting daily functioning and mental health [23,25]. It is estimated that one in five older adults without dementia experience mental health problems [8,9,10,23]. Such conditions, notably anxiety and depression, pose a significant challenge to the pursuit of healthy ageing-in-place, thereby exerting additional strain on already stretched social services and healthcare systems [2,3].

Unfortunately, even though older adults need to restore resources, the restorative pathway has, until now, not been extensively studied with this older target group [17,26,27,28]. Many experimental studies are restricted to younger study populations, such as university students (e.g., [29,30,31,32,33]). However, exploring the restoration process and its applicability and effectiveness concerning ageing populations in current ageing societies has become imperative. Therefore, in this study, we want to review the literature on older adults’ restoration process, elucidate and expand the psychological restoration theory for ageing populations and comprehend influential factors supporting ageing-in-place. Expanding knowledge on restoration theory for older populations can raise awareness about how living environments influence older adults’ mental health and that stress and attention fatigue are potential health risks for ageing populations that hinder ageing-in-place [34,35,36]. The restoration of cognitive and affective resources needs to be considered when discussing holistic and integrative approaches for developing healthy age-inclusive neighbourhoods that suit the capabilities and needs of older adults, including their need for psychological restoration. Furthermore, knowledge about the third pathway can complement the available knowledge of the other two pathways, letting us better understand older adults’ person–environment adaptation process and improving ageing-in-place strategies in policy and design.

A scoping literature review was performed to gain an overview of the existing knowledge about the restorative process for ageing populations and learn more about the influential factors of the restoration pathway. Compared with a systematic review, in a scoping review, qualitative and quantitative studies with a wide variety of methods from different research fields can be included in the review [37,38]. The scoping review method allows to review emerging studies from various fields, using varied study designs in this developing field.

### The General Theory Framework of Psychological Restoration

To expand the psychological restoration theory in the context of older populations, we first need to shed light on the current state of knowledge in the field. The restoration pathway describes the process of psychological restoration: “*the experience of a psychological and/or physiological recovery process that is triggered by particular environments and environmental configurations*” [39] (p. 58). In the environmental psychology literature, this process is explained by two major theories, namely the Attention Restoration Theory (ART) [40] and the Stress Reduction Theory (SRT) [41,42]. Both theories propose that specifically designed environments can trigger cognitive and affective processes, allowing psychological, physiological and social resources to be restored [42,43]. The ART focuses on the capacity to direct attention, a more cognitive process [43], and the SRT focuses on reducing stress and negative moods, a more affective process [42]. Despite these differences, both theories are often used simultaneously to research the mental health effects of restorative environments [44].

Hartig [14] set up a general framework for both restorative theories (Table 1). This framework shows the resources that come into play, get depleted and need restoration, as well as which features of a person–environment transaction can permit and promote the restoration process and the outcomes of the restoration process [14]. Currently, this framework is often deployed in studies with younger target groups like university students (e.g., [29,30,31,32,33]). In this scoping review, we want to elucidate the current framework and explore the applicability and effectiveness of the framework concerning ageing populations. In addition, we focus on studying the features of person–environment transactions of the current general restoration framework that permit and promote restoration (Table 1). These features are environmental requirements of the restoration process through which the depleted resource(s) can be restored. We examine if these features are also relevant for older populations and if additional features are needed to fully describe the restoration process of older populations.

## 2. Methods

### 2.1. Study Design

The scoping review was carried out according to the five-step approach defined by Arksey and O’Malley [37] and adjusted by Levac [38]. The first step, identifying the research question, is presented in the introduction. The four sequential steps are described below. All authors discussed procedures to ensure consistent search methodology, and the PRISMA-ScR checklist was followed to ensure clarity of reporting [45].

### 2.2. Study Identification

A comprehensive search of the literature was conducted in January 2023 to answer the research question: “To what extent is the current framework of psychological restoration theory applicable to older adults, and how do the various elements of this framework contribute to supporting the concept of ageing-in-place?”. The databases searched to identify studies were Scopus, Pubmed, Google Scholar and Web of Science. Search terms were arranged according to the two key themes of the research question: the psychological restoration process and older populations (Table 2). The terms were used to make several search strings. Each string used at least one of the key themes’ search terms, for example: (“psychological restoration”) AND (“age differences” AND “life course”). Furthermore, we tried to find additional papers, book chapters or conference papers by scanning publication lists of well-known authors to prevent publication bias. Additionally, we looked at the backlog of essential journals in the field like the *Journal of Environmental Psychology*, *Environment and Behavior* and the *International Journal of Environmental Research and Public Health*.

### 2.3. Screening and Study Selection

In total, 1083 records were identified by the first author (Figure 1). After removing duplicates, 979 papers were nominated for title selection. First, we removed non-English titles and nonoriginal works, not-peer-reviewed works, theses, and reports, keeping 698 titles for further selection. Hereafter, iterative selection cycles were performed with all three authors. In the first selection cycle, titles were selected by the first author based on two inclusion criteria: older adults and psychological restoration. Doubtful cases were discussed among all three authors. Hereafter, the abstract selection was performed. At the start of the abstract selection, the first author used a random number generator to select 25 titles. Then, inclusion/exclusion decisions were discussed and agreed upon among all three authors. After that, abstracts were selected based on two inclusion criteria: (1) The sample included older adults of 60+ years. Although in gerontology and elsewhere, there are debates about which age constitutes the start of old age, we chose the age of 60, as in several countries people start retiring at this age (for example, in China) [46]. Especially for women, it is a common retirement age worldwide, for example, in Austria, Chilli and Poland [46]. Retirement is a relevant marker of old age for this paper, as the roles people play in society and the daily activities of people are markedly different after retirement. (2) Psychological restoration needed to be discussed following the ART or SRT theory following the general restorative theory framework [14]. Therefore, other forms of restoration were excluded from the study (e.g., building, dental and nature area restoration). Eventually, 55 papers were included for full paper analyses. During the full paper analyses, participant samples were examined by the first author. Papers where all participants were 60 years or older or papers that explicitly looked at age differences within a participant sample, for example, comparing older adults with younger adults or teenagers, were included in the study. Eventually, 39 papers were included in the review (Figure 1).

### 2.4. Data Charting

Following the scoping literature review method [37,38], the next step was setting up a data charting table (Appendix A). Papers were grouped according to the data charting table: author, year, country, research type, theory background, research methods, psychological restoration measures, other measures, participant number and age, other sample characteristics and type of environment. These data were used for descriptive and comparative paper analyses [37,48].

### 2.5. Collation, Summarising and Analysis

After descriptive and comparative paper analyses using the data charting table, the next step was to upload all the selected papers in Atlas.ti for qualitative content analyses. As stated by Smit and Sherman [49], a scoping literature review is a utilised form of qualitative research and Atlas.ti is a suitable application to conduct the paper analyses in a structured way. During the analysis, 105 codes emerged. The codes were grouped following the research questions into three theme groups: (1) features of person–environment transactions that permit the restoration process of older adults, (2) features of person–environment transactions that promote the restoration process of older adults, and (3) contextualising data (e.g., type of environment, demographic data and research limitations). Finally, the code groups were analysed, and the results are presented in the next section.

## 3. Results

### 3.1. Descriptive Results

In this scoping review, thirty-nine peer-reviewed papers were included and analysed to elucidate and expand the restorative theory framework and comprehend influential factors that support ageing-in-place. First, the descriptive results (e.g., methods used and participant groups included) are presented to indicate the quality of the studies included in this review.

The thirty-nine peer-reviewed papers included in this study were published in twenty-six journals from various fields, like gerontology, design, landscape research and environmental psychology. These results indicate the broad distribution of knowledge about the psychological restoration process and old age in various research fields. The scoping review method allowed us to bring this scattered research together, although overall numbers remained small. This scoping review found only one paper published before 2004, written by Travis and McAuley [50] (Figure 2). From 2004 onwards, multiple authors started publishing studies focussing on the restorative experience of older adults, and a slow **increase in publications** over the years can be seen, indicating a growing interest in the psychological restoration process of older adults. This could possibly be related to the rising pressure of ageing populations on healthcare systems and societies and the growing need to mitigate these problems.

The majority of the analysed papers were empirical studies (N = 36). The remaining three studies of the sample were literature reviews. The topic of psychological restoration for older adults is developing, which is reflected in the **wide variety of methods** used (Appendix A). The studies varied between using qualitative (N = 7), quantitative (N = 25) and mixed methods (N = 7) to measure psychological restoration. Which methods were deployed depended on whether authors followed the ART or SRT and if they were more qualitative or quantitatively oriented. Most of the studies were adherent to the ART (N = 21). These studies often used attention tests (e.g., Digit Span Test) to measure restoration or validated questionnaires like the Perceived Restoration Scale (PRS) and the Restoration Outcome Scale (ROS). If the authors were qualitative-oriented, they often analysed interview or spatial data using the four ART features (extent, fascination, being away and compatibility). Studies adherent to the SRT (N = 7) looked at changes in stress levels using self-rated stress scales or measuring physiological characteristics (e.g., blood pressure and heart rate). Eleven studies used methods from both theories. As a result of these varied methods, there are also variations in the presented results. Sometimes, authors present specific features permitting or promoting psychological restoration. Other studies present restorative experiences or design solutions. Because of these variations, the results of different studies cannot always be compared easily. We took these differences between studies into account during our analyses and reported on them further in the following sections.

Although there are many differences between the methods used in the analysed studies, there is also one apparent similarity: **the participant samples** are predominantly comprised of healthy, relatively young (M_age_ = 70.9 years) individuals living independently in the community. Often, participants needed to pass a Mini-Mental State Exam and needed to be able to walk without a walking aid. Only five studies included more vulnerable older adults, including people living in an institution, sitting in a wheelchair, having dementia or recovering from a hip fracture (e.g., [50,51,52]). On the one hand, it is logical for researchers to focus on younger and fit older adults so that there are not too many differences in the study participants. On the other hand, the studies only show results from one specific group, not representing the diversity present in the older population.

The last noteworthy observation about the analysed studies is **the type of environments researched**. Most studies focused on traditional restorative environments like forests and urban parks [20]. However, these environments are not always easily accessible to the older population. Only seven of the thirty-nine studies researched more accessible restorative urban environments close to older adults’ homes, like streets and neighbourhood open spaces (e.g., [20,51,53,54,55]).

This review delves into a relatively underexplored area within the restoration literature: older adults’ restoration process. Despite the fact that older adults have not yet received extensive research attention, we see a slight rise in publications on the topic, indicating a growing interest in the research field. However, it is crucial to acknowledge that research on the psychological restoration of older adults is still in its developmental stages. Studies are dispersed across various disciplines. There are ongoing discussions about methodological issues, a unilateral participant group’s involvement, and there is still little variety in the type of restorative environment studied. More attention should be paid to these issues in future research to further develop the research field of psychological restoration for older populations.

### 3.2. The General Restorative Theory Framework

After the descriptive analysis, we explored if the ten environmental features named in the general restorative theory framework (Table 1) that permit and promote restoration are applicable and effective in the context of an ageing population.

#### 3.2.1. Features That Permit Restoration for Older Populations

The general theory framework proposes three environmental features that permit restoration [15,40,41] (Table 1). These features allow an environment to be free of demands that cause the need for restoration. The SRT states that the **absence of threat** is an essential feature of a restorative environment [42]; one must feel safe before the restoration process can occur. In different qualitative studies, for example, by Jansen [56] and Finlay and colleagues [55], the absence of threat is often described by older adults as an important factor when talking about their restorative experiences (e.g., [55,56]). If older adults feel safe and comfortable in an environment, there is a higher chance that psychological restoration can occur [57,58]. Among others, Qiu and colleagues [59] and Li and colleagues [60] found in their quantitative studies evidence that if spaces can feel unsafe, for example, because of high-density vegetation, feelings of enclosure, insufficient light or too much traffic, this may increase stress and mental fatigue for older adults due to feelings of insecurity [55,59,60,61]. Furthermore, Cassarino and colleagues [62] and Lu and colleagues [63] found evidence in their experimental studies that a good balance between prospect and refuge in an environment increases a sense of security [62,63]. 

ART proposes that experiences of being away and compatibility permit restoration [43] (Table 1). These features include the ability to break with routines, get away from daily life and the ability of the environment to match a person’s capabilities to not further tax “already” depleted resources [14,15,44]. The feature **compatibility** is described in the literature as an essential feature of older adults’ restoration process [22,52,64,65] (Table 3). Scopelliti and Giuliani [64] state, for example, *“a general result claims for the importance of perceived compatibility between elderly persons’ needs and environmental characteristics; when lacking, the consequence is often a dramatic decrease in the restorative potential of everyday settings”* (p. 223). The importance of compatibility for older adults’ restoration process is also evident in other studies, for example, in Fumagalli and colleagues’ [22] analysis of older adults’ descriptions of restorative experiences and in the experimental study of Ottosson and Grahn [52] measuring changes in attention levels in nursing home residents. The importance of the factor compatibility is linked to older adults changing capabilities, which increases the chance of a person–environment misfit due to a lack of compatibility between the person and the environment [17,22]. A lack of compatibility decreases the restorative potential of environments for older adults. Especially in urban, manmade environments, the lack of compatibility is often the main factor negatively affecting perceived restoration [64]. A factor that can negatively impact compatibility is the accessibility of an environment. No restoration can occur if an older adult cannot access or explore the environment [56,66]. Authors like Moore [61] and Marques and colleagues [66] suggest improving accessibility and, thereby, restoration for older adults by designing restorative environments close to the homes of older adults that have toilets, seats, smooth pavement, not much traffic, tree cover, shade and water features [22,61,66].

Opinions about the importance of the permitting feature **being away** from older populations are divided. In different qualitative studies, older adults describe feelings of being away when describing their restorative experiences [22,55,57,67]. Furthermore, in their experimental studies, Rosenbaum and colleagues [53,54] found that environments like senior cafés or senior centres can offer escape experiences for older adults as a home away from home. However, other studies show that being away was significantly less important for older adults compared with younger age groups [26,65]. A possible explanation for this is that older adults may form strong attachments to specific, familiar environments in which they feel safe and comfortable and have strong memories connected. Therefore, they have less need to distance themselves from these environments to gain restoration [26]. Another explanation could be that older adults have less need to distance themselves from ordinary aspects of life to gain restoration [65]. More research is needed to obtain better insight into this feature’s role in permitting psychological restoration for older populations.

**Table 3 ijerph-20-06801-t003:** Features that permit and promote the restoration process of older populations according to the general framework and additional features that can support the restoration process of ageing populations.

Theory	Features of P-E Transactions That Permit Restoration for the Older Population	Features of P-E Transactions That Promote Restoration for the Older Population
**Stress Reduction Theory (SRT)**	**Absence of uncontrollable threat** 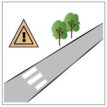	An essential feature of older adults’ restoration process. If older adults feel unsafe in an environment, restoration cannot occur [51,54,55,58,59,60,61,63,64,65,66].	**Perception of natural contents** 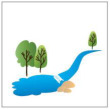	Scenes with water support feelings of calmness and relaxation due to sensory stimulation, also for the older population [22,28,55,59,61,63,64,68].
**Visual stimulus attributes** 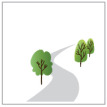	Deflected vistas can enhance curiosity and motivate older adults to go outdoors and explore their everyday environments. There needs to be a right balance of prospect and refuge. Ability to see the environment without feeling exposed [26,50,59,60,61,62,63,65,69].
		**Moderate levels of complexity** 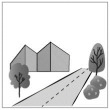	Not named in the reviewed literature in the context of the older population.
		**Gross structure** 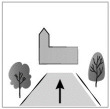	Not named in the reviewed literature in the context of the older population.
**Attention Restoration Theory (ART)**	**Being away** 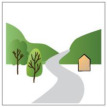	Doubt arises in which way the component being-away is essential for older adults’ restoration process. Escape from every day routines is challenged by the need for social interaction [17,22,26,27,54,55,56,57,58,59,61,67,70,71].	**Fascination** 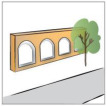	An important feature of older adults’ restoration process. Encourages older adults to explore their surroundings. Authors propose that fascination for older adults is not stimulated by ‘newness’ but by experiencing the familiar in a new way [17,20,22,26,52,53,54,57,58,59,61,64,65,69,70,72,73].
**Compatibility** 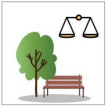	Due to changing capabilities, this feature becomes essential for the restoration process of the older population. Lack of compatibility between the person and the environment dramatically decreases the restorative potential for older adults. Aspects of accessibility play an important role in this feature [17,20,22,26,50,52,54,55,57,58,59,61,64,65,69,70,74].	**Extent** 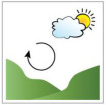	Linked to the presence of (childhood) memories and sensory stimulation. Not named as a condition that will change for the older population [17,20,22,26,57,58,59,61,64,65].
	**Compatibility** 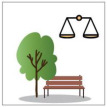	An essential feature for the restoration process of older adults. Although their capabilities change, the environment should enable their life activities [22,54,55,58,59,61,64,65,69,70].
**Outside conventional theories**	**Being with** 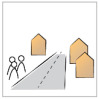	Being-with others is suggested as an essential feature of the restoration process of older adults; however, individual needs need to be taken into account [17,20,22,50,51,52,54,55,56,59,60,61,63,64,65,67,71,75,76,77].	**Familiarity** 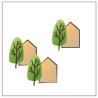	Familiarity can be an additional feature of older adults’ restoration process. Familiar environments can enhance feelings of safety and comfort, promoting the restoration process. A balance between new and familiar elements is important to prevent over or under-stimulation [20,52,55,57,62,64,65,66,74].

#### 3.2.2. Features That Promote Restoration for Older Populations

In addition to features that can permit restoration, the general restorative theory framework also proposes seven features that promote the restoration process [15,40,41] (Table 1). These features can draw a person’s thoughts away from demand and attract and hold their attention, prolonging the restorative process [15,42,43]. Most of these features are also described when researching the restoration process of the older population.

The SRT proposes four features, of which **gross structure** and **moderate complexity** are not or hardly discussed in the context of older populations. However, this does not mean that these are not essential features, only that they have not been researched in the currently reviewed literature. The features **presence of water** and **deflected vista** are studied in the context of older populations and are confirmed as factors that play a role in their psychological restoration process (e.g., [59,75]) (Table 3). Finlay and colleagues found in their qualitative study that water enhances feelings of being connected to nature and is linked to feelings of tranquillity, calmness and spirituality [22,55]. Furthermore, water features such as fountains can enhance curiosity and stimulate the senses (e.g., sounds), promoting restoration for older populations, as found in the empirical studies of Moore [61] and Fumagalli and colleagues [22]. Similarly, Roe and Roe found in their literature review that feature-deflected vistas can enhance curiosity and motivate older adults to go outdoors and explore their everyday environments [17]. Environments with no open exposed view but deflected vistas that offer shelter and provoke exploration of the environment are the most optimal for the psychological restoration of older adults [52,59,63]. However, even though both features are important for the restoration process of older adults, no differences with other age groups are described in the analysed literature.

The ART theory proposes three features of person–environment transactions that promote restoration (Table 1). The feature **extent** is not named in the analysed literature as a feature that will impact the restoration process of older adults differently from other age groups. Some studies link this factor with the presence of (childhood) memories [26,50,57]. Recalling images and emotions that belong to carefree and cheerful moments from life could potentially affect feelings of connectedness (extent) and the restoration process for older adults [22]; however, more research is needed to confirm these assumptions. Furthermore, the literature names the feature **fascination** as an extra important feature for older adults [22,65,73] (Table 3). Jang and colleagues [73] found in their quantitative study that older adults of 60+ years focused on feature fascination (e.g., exploring the surroundings) when judging restorative environments, while younger people focused more on legality and coherence (e.g., structure and orientation), which are part of the feature extent. Others like Liao and colleagues [57] found significant evidence that elements like shade, biodiversity, colourful flowers and vegetation density can enhance fascination in an environment for older populations [17,57,61]. Lastly, feature **compatibility** is named as a feature for permitting and promoting restoration. In addition to allowing restoration, an environment must enable people to carry out activities they want to perform and not limit their abilities, promoting restorative experiences [14]. This is also true for ageing populations; the environment must enable older adults to carry out activities they want to perform that can offer restoration, like exploring the environment, viewing scenery and having fun [64,65]. However, whether this differs from other age groups remains to be studied.

Thus, after a thorough analysis of the literature, it becomes evident that adjustments are needed to adapt the general restorative theory framework to suit ageing populations’ needs. As seen in Table 3, most but not all features of the general restoration theory framework play a role in the context of the older population. Based on the reviewed literature, we suggest prioritising the permitting features. The features safety and compatibility become essential for older adults’ restoration process. The promoting features proposed by the general framework seem similar for older adults compared to other age groups, with the exception of feature fascination, which holds greater significance in older adults’ restoration process compared with other promoting elements outlined in the general framework. Although more research into these permitting and promoting features is needed, these results show that adjustments to the general restorative theory framework are needed to better suit ageing populations’ needs and support healthy ageing-in-place.

### 3.3. Additional Features for the General Restorative Theory Framework

In addition to the general restorative theory framework’s current features, we propose to extend the current framework with two additional features. The first suggestion for an addition is the feature **being with others** (Table 3). Chen and Yuan [75] found in their experimental study that social contact mediated restoration for older adults. Also, Lu and colleagues [63] indicated that companionship significantly impacted the Restoration Outcome Scale. This aligns with others who found that social interaction increases restorative potential, especially in urban environments [59,64,71,78]. A possible explanation is that being together enhances the features of safety and compatibility, increasing the chance that restoration can take place. Another explanation could be that being with others enhances feelings of being part of the world, distancing older adults from their more socially isolated home situation and promoting the feature of being away [17,54]. However, these explanations are not yet confirmed. Scopelliti and Giuliani [65] found in their mixed-method study interesting evidence that being with somebody significantly impacted the restoration process of teenagers and adults but not for older participants. With these varying results, it is essential to remember that social interaction does not always positively affect restoring resources. It depends on the situation and the person. For example, feelings of loneliness, as well as crowding, can negatively influence the restoration process [22,60]. The effects of the social context on restorative processes may vary for different individuals. For example, spending time with family could offer restoration, but for others, it can be mentally and physically draining [56]. Furthermore, the impact of the social component could also be different in natural environments compared with urban environments because social obligations can negatively affect the restoration process [64,65]. More research is needed on how individual differences and social contexts affect older adults’ restoration processes.

Secondly, we suggest adding the feature **familiarity** to the general restorative theory framework when using it in the context of ageing populations (e.g., [62,74]) (Table 3). Berto [74] found in an experimental study a correlation between restoration and familiarity for ageing populations but not for younger age groups. Ottosson and Grahn [52] found in their experiment that older adults have a greater need for familiar surroundings. Familiar environments can enhance feelings of safety and comfort and thereby promote the restoration process [52,55]. Furthermore, Roe and Roe [17] propose that fascination for older adults is not determined by the “newness” of an environment but by experiencing the familiar in a new way. However, a balance must be established between familiar and new elements [52], complementing the feature fascination. Too many new things can create feelings of insecurity, negatively influencing the restoration process, and too many familiar things can undermine the factor fascination and cause understimulation [52,55]. Older adults are more sensitive to this balance than younger people, and this needs to be considered when further studying feature familiarity and its effect on older adults’ restoration processes.

In conclusion, when elucidating the general restoration theory framework in the context of ageing populations, it becomes evident that adjustments and enhancements are needed to adapt the general restorative theory framework to better suit the needs of ageing populations. Based on the reviewed literature, we suggest prioritising safety and compatibility alongside promoting fascination because they hold greater significance in older adults’ restoration processes compared with other elements outlined in the general framework. Moreover, we propose to expand the framework with the features of being with and familiarity; adding these features could improve the framework’s applicability to ageing populations. Although more research is needed, these results must be considered when using the restorative theory framework to develop older adults’ restorative environments, which can improve older adults’ mental health and support ageing-in-place.

## 4. Discussion

The results presented here provide insight into the degree to which the current overarching framework of restorative theory contributes to the understanding of older adults’ psychological restoration process. The inclusion of only thirty-nine studies in this scoping review may seem relatively modest, especially considering that we cast a wide net in terms of the time period and did not select on methodological approaches. When analysing the publication dates, it becomes clear that the topic of psychological restoration among older adults has only recently gained increased attention, and interest in this research area is gradually gaining momentum. It is evident that the research field is in a state of continuous development, as demonstrated by the wide range of literature and methods employed to assess restoration. Furthermore, while attention tests and physiological measures are commonly employed, their suitability for measuring restoration in older populations is subject to debate. For instance, the measurement of heart rate variability presents challenges due to distinct patterns observed in older adults compared with their younger counterparts [79]. Moreover, the study of Cassarino and colleagues [62] found that older adults consistently exhibit lower performance on attention tasks than younger participants, thereby influencing restoration outcome levels. Consequently, we recommend that future studies adopt a mixed-method approach, integrating physiological and psychological measures with participants’ verbal accounts [29]. In a developing research area, the application of diverse methods can offer a range of valuable insights. Nonetheless, in order to improve the comparability of results, it is advisable to consider standardising methods in further studies, particularly in the context of studying psychological restoration in the older population (Table 4).

Another notable methodological issue in the reviewed studies is the homogeneous participant groups commonly employed. Most studies tend to treat older adults as a monolithic entity despite the evident diversity within ageing populations. In reality, older populations encompass a broad spectrum of characteristics and experiences, making it crucial to delve deeper into these variations and how they influence the restoration process. By gaining a nuanced understanding of these distinctions, we can design restorative environments that suit this target group’s different needs and capabilities [75]. To date, it is, for example, unclear if the restoration mechanism works similarly for older adults with cognitive impairments, such as dementia, or those facing physical constraints [57]. Additionally, factors like socioeconomic status, living situation or cultural differences could also be interesting for future research. Only three studies in the review touched on these topics, and while no significant results have emerged thus far, there are indications that these factors could potentially impact the restoration process [67,73]. Therefore, it is imperative to delve deeper into these individual differences and their effects on the restoration process. This knowledge would enable us to provide valuable guidance to researchers, policymakers and designers, allowing them to create environments that can proactively anticipate and adapt to the diverse personal needs of older adults [80] (Table 4).

As mentioned before, this paper set out to elucidate the current restorative theory framework and expand it with two additional features to better suit the restoration process of the older population. The literature showed that the features of person–environment transactions that permit restoration are of extra importance for the restoration processes of older adults, especially the features **absence of threat** and **compatibility**. This is connected to older adults changing capabilities related to the ageing process that increase the chance of a person–environment misfit due to a lack of compatibility between the person and the environment [17,22]. Constantly adapting their behaviour and activities to fit their environment can cost much attention and provoke stress. Therefore, enhancing the compatibility between older populations and their environments can offer psychological restoration benefits [22,52,64,65]. We suggest for future research that these features should be closely monitored, as they can influence the restorative experiences of older adults. Furthermore, similarities could be examined between the restoration process and the person–environment fit model of Lawton [7,81,82], where comparable person–environment transactions are important in the design of environments for older adults. Knowledge from this model may contribute to developing theories for the psychological restoration process for older populations.

Furthermore, our comprehensive literature review shows the significance of **fascination** as an important feature in promoting restoration among ageing populations. Fascinating elements can encourage curiosity and exploration of the living environment. However, a balance between fascinating elements is crucial to the older population. Overstimulation, feelings of unsafety and discomfort can have a negative effect on the restoration processes of older adults [52,55]. Considering this, we suggest the inclusion of the feature of **familiarity** in the general restoration theory framework for older populations (e.g., [63,72]). Familiar environments can lead to a sense of safety and comfort, and experiencing familiar environments in a new way can still encourage curiosity and exploration, promoting the restoration process [17,52,55]. However, it should be noted that the precise impact of the feature familiarity on older adults’ restoration processes needs further research. Theories about lifespan developmental approaches can potentially help with further developing the restorative theory framework [83,84]. These developmental theories explore how earlier life experiences shape people’s lives and reactions as they age. Such theories could potentially elucidate the importance of factors like “familiarity” and shed light on differences in the significance of other elements, such as “being away”.

Lastly, we propose the inclusion of the “being with” feature within the general restorative theory framework. Older adults experiencing stress or attention deficits often tend to isolate themselves, making them more susceptible to stress and initiating a downward spiral that negatively impacts their mental health [23]. Being with people could be essential to reduce loneliness and promote restoration for older populations [17,75]. However, too many (unknown) people can negatively influence the restoration process [56,65]. Therefore, future research should delve deeper into the impact of other people’s presence on the restoration process among older adults (Table 4). Theories about social engagement can potentially give insights into lifespan differences in the social needs of older populations compared with younger groups and could potentially expand the current restoration theory [84,85].

In conclusion, the results of our comprehensive literature review show that not all features of the general restorative theory framework are equally important for older populations. Safety, compatibility and fascination emerge as particularly important for this target group. Furthermore, based on the literature analysis, we propose the inclusion of two additional features to expand the framework: “familiarity” and “being with”. These factors could prove to be crucial determinants in the restoration process of older adults. These findings must be included in future research studies and when developing restorative environments for older populations. By putting more emphasis on these features (safety, compatibility, fascination, familiarity and being with) when developing restorative environments, environments can be created that match older adults’ needs and capabilities regarding restoring resources. Such an approach not only benefits their mental health but also supports the concept of ageing-in-place.

### Strengths and Limitations of the Study

The scoping review methodology exhibits a notable strength in its ability to provide a comprehensive overview of the literature. This makes it particularly well-suited for synthesising research from diverse fields with varying research methods but centred on a common theme [38]. To bolster the credibility of this interdisciplinary study, an extensive search strategy was deployed across multiple databases without imposing date restrictions, and the study identification and selection process underwent a rigorous double review.

As previously highlighted, this review delves into a relatively underexplored area within the literature pertaining to the restorative pathway. Specifically, it addresses the restoration process among older adults, a demographic that has not received extensive research attention. Only thirty-nine studies were identified that examined the restoration process in the context of older populations. Future research should place increased emphasis on this demographic, particularly because disparities in the effectiveness of the theoretical framework were discerned between older adults and younger age groups. Furthermore, when older adults were included in studies, they predominantly comprised healthy individuals from developed nations. Future research endeavours could benefit from a more nuanced examination of individual differences and capabilities within this demographic. From the existing literature, it remains inconclusive whether the factors suggested to enhance the restoration process for older adults are similarly effective for other subgroups of older individuals, such as those dealing with dementia or physical health issues. Consequently, further research is essential to assess the generalizability of the results.

Moreover, it is worth acknowledging that this literature search was conducted exclusively in English, potentially resulting in the omission of evidence from developing countries. Another aspect for consideration is that, due to the scoping review method, no selection was made based on the research methods employed. Consequently, significant methodological variations are evident among the included studies. To enhance the comparability of findings in future research, a standardisation of methods could prove beneficial in the exploration of psychological restoration among older populations.

Lastly, it is vital to recognise that no limitations were imposed on the types of environments investigated. Nevertheless, it is noteworthy that a majority of the studies were conducted in environments that might pose challenges for older adults to access, such as forests and nature parks [10]. To design restorative spaces conducive to enhancing the mental well-being of older adults and supporting ageing-in-place, future studies could explore the restorative characteristics of (semi) public spaces in proximity to the residences of older individuals [20,33,35]. Furthermore, it has become evident that not all restorative factors hold equal significance in every environmental setting. Research by Scopelliti and Giuliani [64] revealed that compatibility and fascination were pivotal in coastal environments, while for urban parks, compatibility and being away were of greater importance to older populations. Consequently, it is conceivable that restoration may manifest through distinct processes in different settings, with varying restorative features playing pivotal roles [64]. Consequently, future research should pay heed to alterations in the restoration process owing to differing individual needs and capabilities, including those stemming from ageing, and align these with the type and design of restorative environments to optimise psychological restoration processes for older populations.

In summary, our understanding of older adults’ psychological restoration processes is steadily expanding, although further research remains imperative. Preliminary findings indicate the necessity for adaptations to the general theoretical framework underpinning restorative environments when considering older populations. This accumulating knowledge can be harnessed to inform the development of restorative environments that promote the mental well-being of older adults and facilitate ageing-in-place.

## 5. Conclusions

This scoping review encompasses a wide range of studies, aiming to provide a comprehensive and detailed overview of the existing knowledge regarding ageing populations’ restoration processes. Upon thorough analysis of the literature, it becomes evident that adjustments and enhancements are needed to adapt the general restorative theory framework to suit the needs of ageing populations better. Based on the reviewed literature, we suggest prioritising safety and compatibility alongside promoting fascination because they hold greater significance in older adults’ restoration processes compared with other elements outlined in the general framework. Moreover, we propose to expand the framework to include concepts like “being with” and “familiarity” to better align with the psychological restoration processes of older populations. These findings should be taken into account when designing restorative environments tailored to older adults. However, it is crucial to acknowledge that research on the psychological restoration of older adults is still in its developmental stages. Studies are dispersed across various disciplines, and there are ongoing discussions about methodological issues. Further research is imperative to fine-tune the general framework to older populations, especially considering the rapidly growing ageing demographic and its impact on healthcare systems and societies.

In conclusion, we anticipate that the insights furnished by this review will offer valuable support to researchers, policymakers and designers as they strive to create age-inclusive neighbourhoods that align with the capabilities and requirements of older adults, including their need for psychological restoration. The aim is to design environments that not only appeal to older adults but also enable them to restore their mental resources. This awareness campaign underscores the profound influence of our living environments on our mental well-being, highlighting the potential health risks posed by stress and attention fatigue in ageing populations, which can hinder the feasibility of ageing-in-place. The insights garnered from this review can serve as a guiding framework to promote the mental health of older individuals and foster healthy ageing-in-place.

## Figures and Tables

**Figure 1 ijerph-20-06801-f001:**
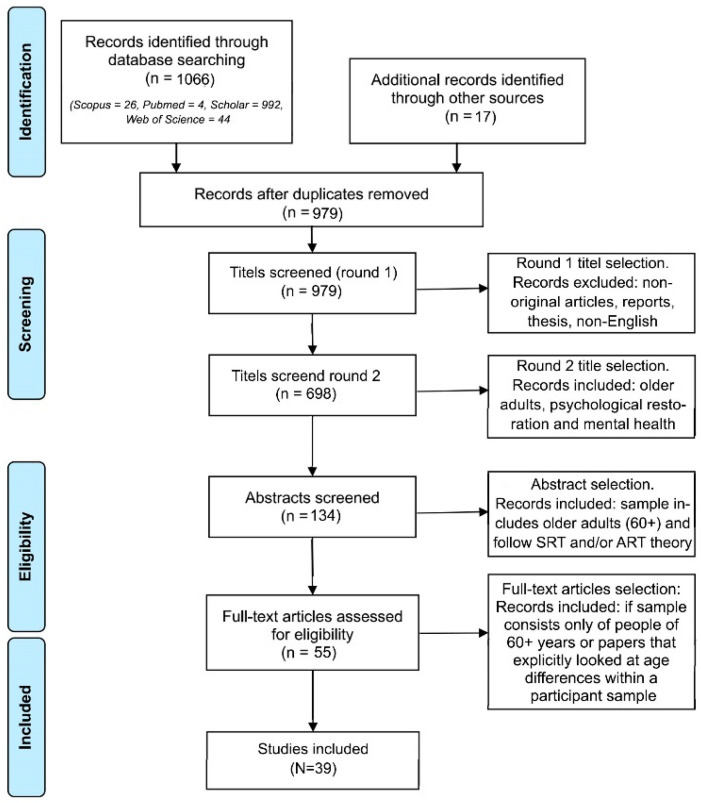
Flow chart of the paper selection process based on PRISMA [47] flow diagram for scoping reviews.

**Figure 2 ijerph-20-06801-f002:**
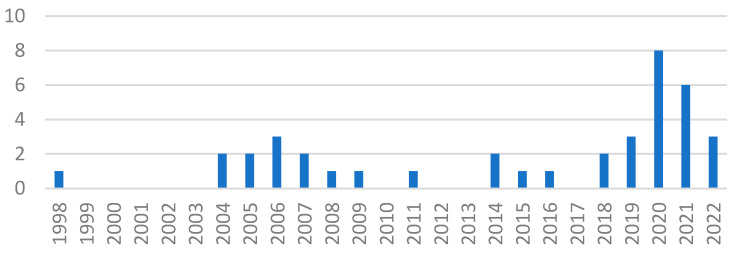
The number of publications per year.

**Table 1 ijerph-20-06801-t001:** A general framework for theories about restorative environments: Stress Reduction Theory and Attention Restoration Theory. The pictograms give additional information about the features of P–E transactions that permit and promote restoration according to the theories. Adapted from Hartig [14] (p. 100).

Theory	Resource Category	Antecedent Condition	Features of P-E Transactions That Permit Restoration		Features of P-E Transactions That Promote Restoration		Outcomes That Can Reflect on Restoration
Stress Reduction Theory (SRT)	Ability to mobilise for action	Psychophysiological stress	**The apparent absence of uncontrollable threat**		**Perception of natural contents**	**Moderate levels of complexity**	More positive self-reported affect, lower blood pressure and cortisol levels
			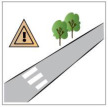		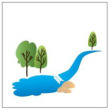	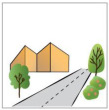	
			In view of a threatful event, feelings of safety need to be encouraged to permit the restoration process.		Scenes with water enhance environmental quality.	Describes the number of separated elements in an environment and the balance between structured and unstructured elements.	
					**Gross structure**	**Other visual stimulus attributes**	
					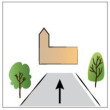	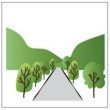	
					The environment needs to give structured information for orientation, for example, a clear focal point.	The line of sight is deflected, hiding what could be lying behind this raises feelings of interest and curiosity. Impacts feelings of spaciousness	
Attention Restoration Theory (ART)	Ability to direct attention	Directed attention fatigue	**Being away**	**Compatibility**	**Fascination**	**Extent**	Improved performance on standardised tests of cognitive abilities
			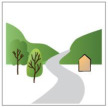	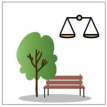	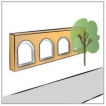	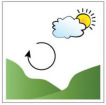	
			Escape (physically or mentally) from everyday routine pressures and obligations.	The perceived fit between the environment and the individual needs and inclinations.	The environment’s capability to involuntarily catch one’s attention and not demand mental effort.	Refers to properties of connectedness. The environment feels like a whole (coherence) and promises to engage one’s mind (scope).	
					**Compatibility**		
					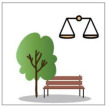		
					The way that an environment enables people to experience restorative activities.		

**Table 2 ijerph-20-06801-t002:** Search items used in the search per key theme.

Psychological restoration	Restoration likelihood; Restorative experiences; Restorative potential; Perceived restoration; Restorative environment; Attention restoration; Stress
Older population	Elderly; Older adult; Third age; Fourth age; Life span; Life course; Old people; Elder; Age differences; Senior; Older individuals

**Table 4 ijerph-20-06801-t004:** Critical areas of recommendation for future research.

Method	Standardisation on psychological restoration measures for older adults.
	Research the compatibility of physiological measures and attention tests on older populations.
Individual and generational differences	More research is needed with a variety of older participants, such as older old individuals, people with cognitive disabilities or different cultural backgrounds.
Features of person-environment transaction	Further research on the permitting and promoting features proposed in the general theory and how they are applicable to older populations.
	Investigation of additional features (e.g., being with and familiarity) and their influence on the psychological restoration process for older adults.
Type of environment	Research the restorative potential of accessible environments close to older adults’ homes (for example, neighbourhood open spaces).

## Data Availability

No new data were created in this literature review. Only existing sources were used for analyses; see references. Data sharing does not apply to this article.

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
