# Peer review of "Elucidating and Expanding the Restorative Theory Framework to Comprehend Influential Factors Supporting Ageing-in-Place: A Scoping Review"

_ijerph, 2023, doi:10.3390/ijerph20186801_

Round 1

Reviewer 1 Report

This is an interesting scoping review designed to apply a psychological restoration framework, that has largely been applied to university students, to those who are aging in place.  They conducted a qualitative analysis of 39 extant papers on the topic of SRT or ART and made some suggestions for extension of the framework. 

There are some suggestions for the authors to consider to improve this manuscript and make it compatible for publication. 

First, there are many grammatical and spelling errors throughout the manuscript. Careful editing is required.

In the introductions, the second sentence does not support the first sentence. The second sentence should provide some reference to prior population as well to give a sense of rapid aging. 

"Due to scientific advances, improved living conditions and medical breakthroughs, 32 our global population is rapidly ageing. In 2050, nearly 2.1 billion people are expected to 33 be 60 years and older [1]."

Next, the authors identify that the literature on restorative theory is scant, yet 39 papers met criteria for review. I would be reluctant to describe this literature as scant. It is a smaller body of work, but it seems that scant may not be the correct word here. It may be a small body of work relative to that found among university students. Perhaps a larger issue to present is that a review of this nature has not been done before with respect to community-dwelling adults.

Methods

2.3 Screening and Study Selection

The authors wrote "The age of 60 was chosen because people start retiring at this age in many countries." Please provide a reference for this statement as well as examples of countries where retirement starts around age 60. 

Results, while qualitative, seems to lack substance. Readers would be interested in really understanding the quality of the literature being reviewed. When the author is making statements regarding the importance of restorative permitting and promoting components, a contextualizing regarding the quality of the literature and type of literature available (i.e., quantitative, qualitative, mixed methods) would be important. Critiques on the rigor seem needed throughout to help the reader understand how to interpret the results. 

Much of the information provided in the results sections seems like it belongs in the discussion (e.g., lines 233-236 on page 7 – “It is vital to learn more . . .” and lines 242-245 – “it could be interesting”)

Discussion:

the authors should consider the inclusion of developmental theories regarding aging as support for understanding why older adults may experience "getting away" differently than younger adults. Can the authors look more broadly at aging theory and developmental literature to support or understand discrepancies with other literature? For example, the authors might draw parallels from developmental literature on social engagement. 

Multiple edits are required throughout the manuscript.

Author Response

[Please see the attachment]

We would like to thank you for your review dated 29th August 2023 and the opportunity to revise our manuscript "Elucidating and expanding the restorative theory framework to comprehend influential factors supporting ageing-in-place: a scoping review". We would also like to take this opportunity to express our thanks for the positive feedback and helpful comments for correction or modification. We sincerely believe that the comments and suggestions have significantly improved this manuscript. Please find enclosed a point-by-point explanation of the changes made in response to the comments and suggestions we received.

Sincerely yours 

Reviewer 2 Report

Thank you so much for this opportunity to review this scoping review on theory-informed factors influencing the restorative process among older adults. This paper is well-written and insightful in general. However, the clarity of the research question, methods, and result presentation can be further improved to highlight its contribution.

1.     p.2 Line 49” restoration perspective –“: Please consider adding one more sentence with a few examples of restorative processes and resources.  The authors later defined and explained the relevant concepts. However, I think it would be helpful to clarify the concept/definition here because this is the first time the term is used.

2.     p.2 line 67 “Unfortunately, even though older adults do need to restore resources, the restorative pathway is, until now, predominantly tested with students.” This is a critical summary and assessment of the existing literature. Please expand, provide some brief examples of existing studies, and cite.

3.     P3-4. Line 115: “This framework is mainly used for studies with student populations.” Please cite and give 1-2 examples from the existing literature. Thanks!

4.     p.4 line 134. “Search terms were based on two key themes of the research question: psychological restoration and the older population (Table 2).” I think it might be helpful to clearly state your research question here in a complete sentence.  What aspect of psychological restoration among the older population are you interested in? Are you looking for factors promoting/hindering restoration or intervention facilitating restoration? Please clarify to avoid confusion.

5.     p.5 “2.4 Data Screening“:How many reviewers are involved in each process? How were conflicts resolved?

6.     p.13 “Strengths and limitations of the study”. Please consider breaking the long paragraph in this section down into two paragraphs for clarity.  Another limitation is that this scoping review does not identify whether all restorative factors matter equally to various subgroups of older adults. In other words, it is unclear what theory and which restorative resource is more effective for which group of older adults (e.g., older adults living with physical disabilities, older adults living with Alzheimer's disease and related dementia, older adults living alone). Whether or not this is a limitation depends on how the authors define their specific research questions for this scoping study. Thank you very much!  

Minor revisions. 

Author Response

[Please see the attachment]

We would like to thank you for your review dated 1st September 2023 and the opportunity to revise our manuscript "Elucidating and expanding the restorative theory framework to comprehend influential factors supporting ageing-in-place: a scoping review". We would also like to take this opportunity to express our thanks for the positive feedback and helpful comments for correction or modification. We sincerely believe that the comments and suggestions have significantly improved this manuscript. Please find enclosed a point-by-point explanation of the changes made in response to the comments and suggestions we received.

Sincerely yours 

Round 2

Reviewer 2 Report

The authors have addressed my comments thoroughly. Thank you!